# Hypoxia Alters the Proteome Profile and Enhances the Angiogenic Potential of Dental Pulp Stem Cell-Derived Exosomes

**DOI:** 10.3390/biom12040575

**Published:** 2022-04-14

**Authors:** Baoyu Li, Xuehong Xian, Xinwei Lin, Luo Huang, Ailin Liang, Hongwei Jiang, Qimei Gong

**Affiliations:** 1Hospital of Stomatology, Sun Yat-Sen University, Guangzhou 510055, China; liby39@mail2.sysu.edu.cn (B.L.); linxw33@mail2.sysu.edu.cn (X.L.); huangl238@mail2.sysu.edu.cn (L.H.); liangailin@mail2.sysu.edu.cn (A.L.); 2Guangdong Provincial Key Laboratory of Stomatology, Guangzhou 510080, China; 3Guanghua School of Stomatology, Sun Yat-Sen University, Guangzhou 510055, China; 4Department of Stomatology, The Sixth Affiliated Hospital, Sun Yat-Sen University, Guangzhou 510655, China; xianxuehong@163.com; 5Foshan Stomatological Hospital, Foshan University, Foshan 528000, China

**Keywords:** hypoxia, dental pulp stem cells, exosomes, angiogenesis, proteomics, lysyl oxidase-like 2

## Abstract

Dental pulp stem cells (DPSCs) and their exosomes (Exos) are effective treatments for regenerative medicine. Hypoxia was confirmed to improve the angiogenic potential of stem cells. However, the angiogenic effect and mechanism of hypoxia-preconditioned DPSC-Exos are poorly understood. We isolated exosomes from DPSCs under normoxia (Nor-Exos) and hypoxia (Hypo-Exos) and added them to human umbilical vein endothelial cells (HUVECs). HUVEC proliferation, migration and angiogenic capacity were assessed by CCK-8, transwell, tube formation assays, qRT-PCR and Western blot. iTRAQ-based proteomics and bioinformatic analysis were performed to investigate proteome profile differences between Nor-Exos and Hypo-Exos. Western blot, immunofluorescence and immunohistochemistry were used to detect the expression of lysyl oxidase-like 2 (LOXL2) in vitro and in vivo. Finally, we silenced LOXL2 in HUVECs and rescued tube formation with Hypo-Exos. Hypo-Exos enhanced HUVEC proliferation, migration and tube formation in vitro superior to Nor-Exos. The proteomics analysis identified 79 proteins with significantly different expression in Hypo-Exos, among which LOXL2 was verified as being upregulated in hypoxia-preconditioned DPSCs, Hypo-Exos, and inflamed dental pulp. Hypo-Exos partially rescued the inhibitory influence of LOXL2 silence on HUVEC tube formation. In conclusion, hypoxia enhanced the angiogenic potential of DPSCs-Exos and partially altered their proteome profile. LOXL2 is likely involved in Hypo-Exos mediated angiogenesis.

## 1. Introduction

Dental pulp is a highly innervated and vascularized tissue surrounded by dentin walls. Reports indicate that 40% of dental pulp volume is comprised by vasculature [1], which is not only responsible for nutrient supply and waste removal but is also involved in most of the biological functions of teeth. However, in most cases, pulp tissue receives blood supply from a narrow apical foramen. Therefore, pulp vascularization is an essential and challenging task in dental pulp regeneration.

Dental pulp stem cells (DPSCs) are mesenchymal stem cells (MSCs) isolated from adult dental pulp. DPSCs are great alternatives for bone marrow mesenchymal stem cells (BMSCs) because they can be obtained from extracted teeth, which causes less invasion. Exosomes (Exos) are 30–150 nm-sized extracellular vesicles (EVs), which contain a diverse array of signaling molecules (e.g., proteins, miRNAs, mRNAs, non-coding RNAs, and lipids) and play a critical role in cellular communication. Exosomes perform biological functions similar to their parent cells and possess a broader clinical application prospect in regenerative medicine due to their non-proliferation and low immunogenicity. As summarized by Maqsood et al., exosomes derived from different sources of MSCs showed different advantages [2]. DPSC-Exos seemed to show unique advantages in pulp regeneration. A recent study found that bioengineered teeth established by a natural decellularized tooth matrix (DTM) combined with DPSC aggregates regenerated both dental pulp and periodontal tissue of avulsed teeth after reimplantation, during which exosomes derived from DPSC aggregates may play an essential role by upregulating the odontogenic and angiogenic ability of DPSCs [3]. Furthermore, an in vitro study confirmed that DPSC-EVs-fibrin gels facilitated the formation of vascular-like structures and the deposition of collagen I, III, and IV in 3D HUVECs and DPSCs co-cultured systems [4]. In terms of angiogenesis, our previous study revealed that dental pulp cell-derived exosomes promoted human umbilical vein endothelial cells (HUVECs) proliferation, proangiogenic factor expression and tube formation [5]. Wu et al. [6] found that exosomes from stem cells from deciduous teeth (SHED) aggregates improved angiogenesis during pulp regeneration via regulating TGF-β/SMAD2/3 signaling. DPSCs and their exosomes are becoming effective therapeutic tools in regenerative medicine.

However, Merckx et al. [7] showed that DPSC-EVs are less potent in endothelial cell chemotaxis and ovo neovascularization compared to BMSC-EVs. It has been reported that changes in the cellular microenvironment could enhance the biological activities of DPSC-Exos. Studies have shown that LPS-pretreated DPSC-Exos demonstrated stronger potential in angiogenesis [8], odontogenic differentiation [9] and pulp regeneration [10]. Zhou et al. [11,12] demonstrated that DPSC-EVs derived from periodontally diseased teeth exerted a more robust potential to stimulate angiogenesis, suggesting that the inflammatory microenvironment could improve the angiogenic property of DPSC-Exos. Inflammation in dental pulp usually produces a hypoxic microenvironment. However, the regulation and mechanism of hypoxia on DPSC-Exos are still unclear.

Hypoxia is considered to be a driving angiogenic force in injured dental pulp tissue [13]. Therefore, comprehension of dental pulp tissue responses to hypoxia is essential for regenerative endodontics and dental traumatology. DPSCs have been shown to promote angiogenesis via a mechanism that is potentiated by hypoxia. Under hypoxic stimulation, the differentiation potential and paracrine action of dental pulp cells were significantly enhanced [14]. Proteomic analysis indicated that DPSCs contain a plethora of angiogenesis-related proteins such as HIF-1α, VEGFA, KDR and TGFβ1 under hypoxic conditions, which contribute to tissue regeneration [15]. Hypoxia-inducible factor 1α (HIF-1α) overexpressing DPSC-derived exosomes [16] have a high angiogenic capacity via the enhanced expression of the Notch ligand Jagged1, which could have potential applications for ischemia-related disease treatment. Considering that hypoxia enhanced the angiogenic potential of DPSCs-conditioned media [17] and exosomes derived from other types of MSCs [18,19,20,21,22], we hypothesized that hypoxia could also enhance the angiogenic potential of DPSC-Exos.

Hypoxia preconditioned DPSC-derived exosomes (Hypo-Exos) have not been characterized in detail and their angiogenic role remains unresolved. In this study, we isolated Hypo-Exos and DPSC-derived exosomes under normoxic conditions (Nor-Exos) and investigated the role of Hypo-Exos in angiogenesis. To study the mechanisms of the angiogenic effect of Hypo-Exos, we performed an isobaric tag for relative and absolute quantitation (iTRAQ)-based proteomics analysis of the two DPSC-Exos. These will promote the application of DPSCs exosomes as a cell-free therapy in regenerative medicine.

## 2. Materials and Methods

### 2.1. Cell Isolation, Culture, and Identification

Human third molars or premolars were harvested with written informed consent from patients (18–25 years of age) undergoing extraction for orthodontic or therapeutic reasons at the *Hospital of Stomatology, Guanghua School of Stomatology, Sun Yat-sen University, Guangzhou*. DPSCs were isolated from dental pulp tissue as described previously [23] from different donors. Cells were cultured at 37 °C in 5% CO_2_ using αMEM (Gibco, Grand Island, NY, USA) supplemented with 10% FBS (Gibco, Grand Island, NY, USA), 100 U/mL penicillin, and 100 mg/mL streptomycin (Sigma, Saint Louis, MO, USA). Culture media were changed every three days. Experiments were performed with DPSCs from passages 3 to 5.

Flow cytometry (BECKMAN, Miami, FL, USA) was conducted to identify stem cell positive markers CD29 (BD Pharmingen, San Diego, CA, USA), CD73 (BD Pharmingen, San Diego, CA, USA), CD90 (BD Pharmingen, San Diego, CA, USA) and CD105 (Abcam, Cambridge, UK) or negative markers CD34 (BD Pharmingen, San Diego, CA, USA), CD45 (BD Pharmingen, San Diego, CA, USA) and HLA-DR (BD Pharmingen, San Diego, CA, USA). The multipotent differentiation of DPSCs into osteoblasts, adipocytes and endothelial cells (ECs) in vitro was also determined. For osteogenic differentiation, DPSCs were induced in Human Dental Pulp Stem Cell Osteogenic Differentiation Medium (Cyagen, Santa Clara, CA, USA) for 21 days and the mineralized nodules were stained with Alizarin Red S (Cyagen, Santa Clara, CA, USA). To identify adipogenic differentiation, DPSCs were cultured in Human Dental Pulp Stem Cell Adipogenic Differentiation Medium (Cyagen, Santa Clara, CA, USA) for 21 days and measured by Oil Red O staining (Cyagen, Santa Clara, CA, USA). For endothelial differentiation, DPSCs were induced in endothelial differentiation medium supplemented with 2% FBS, 50 μg/L VEGF (Peprotech, Rocky Hill, NJ, USA), and 10μg/L Bfgf (Peprotech, Rocky Hill, NJ, USA) for seven days and seeded on Matrigel (10 mg/mL, Corning, Bedford, MA, USA) to form capillary-like structures.

HUVECs were purchased commercially (ScienCell, San Diego, CA, USA), and expanded in endothelial cell medium (ECM; ScienCell, San Diego, CA, USA) containing 5% FBS and 1% endothelial cell growth supplement at 37 °C with 5% CO_2_. Cells below passage 7 were used for further experiments.

### 2.2. Exosome Isolation and Identification

Exosomes were isolated from the culture media of DPSCs at passage 5 cultured under normoxia or hypoxia as previously described [4]. Briefly, DPSCs at passage 5 were cultured in 150 mm culture dishes in αMEM supplemented with 10% FBS. Upon reaching 80% confluence, cells were washed with phosphate-buffered saline (PBS) and cultured in serum-free αMEM for 48 h under 21% O_2_ (normoxia) or 2% O_2_ (hypoxia). The condition culture media were then collected and centrifuged for 10 min at 300× *g* followed by 10 min at 2000× *g* to remove the cells. After centrifugation at 10,000× *g* for 30 min to eliminate the debris, the supernatant was concentrated with an Amicon Ultra-15 Centrifugal Filter Unit (Millipore Corp, Billerica, MA, USA) to an appropriate volume of 4000× *g*. Subsequently, the concentrated supernatant was centrifuged at 100,000× *g* for 1 h. The pellets were resuspended in PBS and centrifuged at 100,000× *g* for 1 h. All procedures were conducted at 4 °C. Exosomes derived from DPSCs under normoxia (Nor-Exos) or hypoxia (Hypo-Exos) were finally resuspended in PBS and stored at −80 °C. The exosome protein concentration was quantified with a BCA Protein Assay Kit (Cwbio, Beijing, China).

Transmission electron microscopy (TEM) was used to characterize the morphology of Nor-Exos and Hypo-Exos. 10 μL of exosome suspension was placed onto formvar/carbon-coated nickel grids. After 1 min at room temperature, the excess liquid was removed, and the samples were stained with uranyl acetate for 1 min. The grids were dried for several minutes and examined using an HT-7700 transmission electron microscope (100 kV, Hitachi, Tokyo, Japan) to characterize the morphology of exosomes. Nano flow cytometry was utilized to determine the size range of exosomes with a NanoFCM N30E (NanoFCM, Xiamen, China). The exosomal markers CD9 (1:1000; Affinity Biosciences, Cincinnati, OH, USA), CD63 (1:1000; Affinity Biosciences, Cincinnati, OH, USA) and CD81 (1:1000; Zen BioScience, Chengdu, China) were examined using Western blot.

### 2.3. Exosome Uptake

The PKH26 fluorescent labeling kit (Umibio, Shanghai, China) was utilized to label exosomes according to the manufacture’s protocol. In brief, exosomes were incubated with 5 μL PKH26 dye solution and 45 μL Dilution C solution for 10 min in a dark environment. 10 mL PBS was added to terminate the labeling reaction, and the exosomes were subsequently harvested and washed by centrifugation (100,000× *g* for 70 min). Following this, HUVECs were cultured with PKH26-labeled exosomes for 6 h followed by 4% paraformaldehyde (PFA) fixation for 15 min. After three washes with PBS, the cytoskeleton was stained by phalloidin (Abbkine, Redlands, CA, USA) for 30 min and nuclei were labeled with DAPI (Beyotime, Shanghai, China) for 5 min. Finally, exosome uptake images were captured using a laser confocal microscope (Olympus, Tokyo, Japan).

### 2.4. Cell Proliferation Assay

A Cell Counting Kit-8 (CCK-8) assay was carried out to assess cell proliferation. HUVECs were seeded onto 96-well plates at a density of 2 × 10^3^ cells/well. After 12 h, 10 μg of Nor-Exos or Hypo-Exos were added per milliliter cell culture media (10 μg/mL). An equal volume of PBS was added to the control group. After one, two, and three days, 10 μL/well CCK-8 solution (Dojindo, Kumamoto, Japan) was added to the culture media and incubated for 2 h at 37 °C. Finally, the absorbance at 450 nm was detected by a microplate reader (Biotek, Winooski, VT, USA).

### 2.5. Cell Migration Assay

Migration of HUVECs was assessed using 24-well 6.5 mm transwell chambers with 8 μm pores (Falcon Corning, Durham, NC, USA). In each lower chamber, 600 μL of ECM was added containing PBS, Nor-Exos or Hypo-Exos (10 μg/mL). HUVECs (1 × 10^4^ in 100 μL of ECM without FBS) were added to the upper chamber and allowed to migrate to the lower part of the chamber for 6 h at 37 °C. Non-migrated cells on the upper surface of the upper chamber were removed with cotton buds. Migrated cells were fixed with 4% PFA for 15 min and stained with 0.5% crystal violet dye. Images were taken with an inverted microscope (Zeiss, Oberkochen, Germany). The number of migrated cells was calculated using Image J (Fiji software version 2.1.0/1.53.c, National Institutes of Health, Bethesda, MD, USA).

### 2.6. Tube Formation Assay

HUVECs were pretreated with PBS, Nor-Exos or Hypo-Exos (10 μg/mL) for 24 h before the tube formation assay. Chilled Matrigel (10 mg/mL, Corning, Bedford, MA, USA) was placed in a 48-well plate (150 µL/well) and allowed to polymerize for 1 h at 37 °C. Pretreated HUVECs were resuspended in culture media with Nor-Exos or Hypo-Exos (10 μg/mL), or an equal volume of PBS as a control. A total of 3 × 10^4^ HUVECs were seeded onto each well precoated with Matrigel and incubated in a 5% CO_2_ incubator at 37 °C. The tube formed after 6 h was observed through an inverted microscope (Zeiss, Oberkochen, Germany). Images were captured and analyzed using Image J software. LOXL2-silenced HUVECs were pretreated with PBS or Hypo-Exos (10 μg/mL) for 24 h and seeded onto Matrigel. After incubation for 6 h, images were captured and analyzed.

### 2.7. Quantitative Real-Time PCR (qRT-PCR) Analysis

Total cell RNA was extracted using RNA-quick Purification Kit (Yishan, Shanghai, China) and reverse transcribed using PrimeScript™ RT reagent (Takara, Tokyo, Japan). Subsequently, qRT-PCR analysis was performed via LightCycler96 (Roche, Basel, Switzerland) with qPCR SYBR Green detection reagent (Yeasen, Shanghai, China). Data were analyzed by formula 2^(-(∆∆CT)) with β-actin as an internal reference. The sequences of the mRNA primers are listed as follows. β-actin: forward, 5′-CATGTACGTTGCTATCCAGGC-3′, and reverse, 5′-CTCCTTAATGTCACGCACGAT-3′. KDR: forward, 5′-TACGTTGGAGCAATCCCTGT-3′, and reverse, 5′-TACACTTTCGCGATGCCAAG-3′. CD31: forward, 5′-CCAAGCCCGAACTGGAATCT-3′, and reverse, 5′-CACTGTCCGACTTTGAGGCT-3′. MMP9: forward, 5′-TCCCTGGAGACCTGAGAACC-3′, and reverse, 5′-CCACCCGAGTGTAACCATAGC-3′. VEGFA: forward, 5′-TTGCTGCTCTACCTCCACCAT-3′, and reverse, 5′-GGTGATGTTGGACTCCTCAGTG-3′. CXCR4: forward, 5′-ATCAGTCTGGACCGCTACCT-3′, and reverse, 5′-CCACCTTTTCAGCCAACAGC-3′. SDF-1: forward, 5′-CTACAGATGCCCATGCCGAT-3′, and reverse, 5′-CAGCCGGGCTACAATCTGAA-3′. LOXL2: forward, 5′-AGGACATTCGGATTCGAGCC-3′, and reverse, 5′-CTTCCTCCGTGAGGCAAAC-3′.

### 2.8. Western Blot Analysis

To analyze protein expression, Western blot analysis was performed. Exosomes samples were resuspended in a RIPA lysis buffer (Beyotime, Shanghai, China) with 1% protease inhibitor cocktail (Cwbio, Beijing, China) and lysed on ice for 30 min. Cell samples in 6 well plates were washed with PBS and lysed in RIPA lysis buffer (Beyotime, Shanghai, China) with 1% protease inhibitor cocktail (Cwbio, Beijing, China) on ice for 30 min. Subsequently, lysates were centrifuged at 12,000× *g* for 20 min, and supernatants were collected. A BCA protein assay kit (Cwbio, Beijing, China) was used to assess protein concentration. SDS-PAGE loading buffer (Cwbio, Beijing, China) was added to the lysates, and samples were boiled at 99 °C for 10 min. Equal amounts of protein were separated by 4–12% SDS-PAGE gels (GenScript, Piscataway, NJ, USA) and transferred to PVDF membranes (Millipore Corp, Billerica, MA, USA). Membranes were blocked with 5% bovine serum albumin in TBST for 1.5 h and incubated overnight with primary antibodies at 4 °C. The primary antibodies were diluted as follows: CD9 and CD63 (1:1000; Affinity Biosciences, Cincinnati, OH, USA), CD81 (1:1000; Zen BioScience, Chengdu, China), KDR (1:1000; Cell Signaling Technology, Danvers, MA, USA), CD31 (1:1500; Cell Signaling Technology, Danvers, MA, USA), MMP9 (1:1000), VEGFA (1:1000; Proteintech, Chicago, IL, USA), CXCR4 (1:1000; Abcam, Cambridge, UK), SDF-1 (1:1000; Abcam, Cambridge, UK), LOXL2 (1:1000; Abcam, Cambridge, UK), HIF-1α (1:1000; Proteintech, Chicago, IL, USA) and β-actin (1:3000; Emar, Beijing, China). Subsequently, the membranes were incubated with an HRP-conjugated anti-rabbit secondary antibody (1:10,000; Emar, Beijing, China) for 1 h at room temperature. Finally, immunocomplexes were visualized with a chemiluminescent HRP substrate (Millipore Corp., Billerica, MA, USA) and detected using a Chemiluminescence Imaging System (Syngene, Cambridge, UK). ImageJ software was utilized to quantify protein expression levels.

### 2.9. Protein Extraction, Digestion and iTRAQ Labeling

DPSCs were isolated and cultured as mentioned earlier, and cells were combined from different donors. Exosomes were isolated from the pooled culture media of DPSCs under normoxia (Nor-Exos) and hypoxia (Hypo-Exos) by ultracentrifugation. The experiments were performed three times. Nor-Exos and Hypo-Exos were collected, and samples corresponding to each group were pooled. The protocol of protein extraction was based on previous studies [24,25], with some modifications. Nor-Exos and Hypo-Exos were suspended in an RIPA lysis buffer with PMSF (Bocai, Shanghai, China) and sonicated on ice for 5 min. After centrifugation for 20 minutes at 4 °C at 25,000× *g*, the supernatant was reduced with 10 mM DTT at 56 °C for 1 h and alkylated with 55 mM IAM in the dark for 45 min. Subsequently, the supernatant was added to five volumes of cold acetone and incubated at −20 °C for 2 h. The precipitates were collected and washed twice with cold acetone. After precipitation with cold acetone, the precipitates were resuspended in 0.5 M TEAB for 15 min. The supernatant was collected and quantified by the Bradford assay according to the manufacturer’s instructions. Proteins were double verified by SDS-PAGE. In brief, 30 μg protein from each sample was mixed with loading buffer and heated at 95 °C for 5 min. After centrifugation at 25,000× *g* for 5 min, 12% SDS-PAGE electrophoresis (constant voltage 120 V, 120 min) was performed and the gel was stained with Coomassie blue. 100 μg protein from each sample was digested with 2.5 μg trypsin gold (Promega, Madison, WI, USA) at 37 °C for 4 h. After adding an additional 2.5 μg trypsin for 8 h, the enzymatic peptide was desalinated with a Strata X C18 column (Phenomenex, Torrance, CA, USA) and vacuum-dried. According to the manufacturer’s protocol, each sample was dissolved in 50 μL isopropanol by vortexing. The peptide sample was dissolved in 0.5 M TEAB and transferred to the corresponding room temperature iTRAQ reagent for incubation at room temperature for 2 h.

### 2.10. Peptide Fractionation, High-Performance Liquid Chromatography (HPLC), and Mass Spectrometry Analysis

Peptide fractionation was performed using an LC-20AB HPLC pump system (Shimadzu, Kyoto, Japan) equipped with a 4.6 mm × 250 mm Gemini C18 column containing 5 µm particles (Phenomenex, Torrance, CA, USA). The iTRAQ labeling peptide mixtures were resuspended in 2 mL buffer A (5% ACN pH 9.8). Afterward, the peptides were eluted at a 1 mL/min flow rate with a gradient of 5% buffer B (95% CAN, pH 9.8) for 10 min, 5–35% buffer B for 40 min, 35–95% buffer B for 1 min, 100% buffer B for 3 min, and 5% buffer B for 10 min. The elution was monitored by 214 nm absorbance and the fractions were collected every 1 min. The collected fractions were eventually combined into 20 fractions and vacuum-dried. Each pool of mixed peptides was then resuspended in buffer A (2% ACN, 0.1% FA) and centrifuged at 20,000× *g* for 10 min. The supernatant was loaded onto a C18 trap column (Phenomenex, Torrance, CA, USA) using an LC-20AD nano-HPLC instrument (Shimadzu, Kyoto, Japan) at a flow rate of 300 nL/min. The elution used a step-wise linear program: 5% buffer B (98% CAN,0.1% FA) for 8 min, 8–35% buffer B for 35 min, 35–60% buffer B for 5 min, 60–80% buffer B for 2 min, 80% buffer B for 5 min, and 5% buffer B for 10 min. LC-MS/MS analysis was performed with an AB SCIEX nano LC-MS/MS (Triple TOF 5600) system. During data collection, the MS analysis parameters were as follows: the ion source voltage was 2.3 kV; the MS1 scan range was 350~1500 Da with a cumulative time of 250 ms. Screening conditions for the secondary fragmentation were as follows. The MS2 m/z scan range was fixed at 350–1250 Da with a cumulative time of 100 ms; charge 2+ to 5+ and the dynamic exclusion time was set to 12 s.

### 2.11. Bioinformatics Analysis

The raw MS/MS data were converted to MGF format and files were searched using the Mascot search engine (Matrix Science, London, UK; version 2.3.02) to identify and quantify proteins. IQuant software was utilized for iTRAQ data quantification. Proteins with an average ratio of more than 1.2 fold and a *p*-value less than 0.05 were determined as differentially expressed proteins (DEPs). All proteins with FDR less than 1% were subjected to analyses including Gene Ontology (GO) and Kyoto Encyclopedia of Genes and Genomes (KEGG) pathway analyses. Based on the DEPs, we further analyzed the results of the GO enrichment and KEGG pathway enrichment analyses. In addition, we also analyzed interactions between the significant KEGG pathways and performed protein-protein interaction (PPI) analysis with STRING (Version 11.0) and Cytoscape software (Version 3.8.2).

### 2.12. Immunofluorescent Analysis

Immunofluorescence staining was conducted to characterize the expression and distribution of HIF-1α and LOXL2 in DPSCs. DPSCs were seeded on coverslips in 24-well plates (1.5 × 10^4^ cells per well) and cultured at 37 °C overnight. The culture media were replaced with serum-free media and cells were cultured under normoxia or hypoxia for 48 h. Cells were fixed with 4% PFA for 20 min and permeabilized with 0.3% Triton X-100 in PBS for 10 min. After blockading in 10% goat serum for 1 h, cells were incubated overnight with primary antibodies against HIF-1α (1:100; Proteintech, Chicago, IL, USA) and LOXL2 (1:100; Abcam, Cambridge, UK) at 4 °C. The next day, after washing with PBS, cells were incubated with fluorescein-conjugated goat anti-mouse antibody (1:200, DyLight 488; Abbkine, Redlands, CA, USA) and goat anti-rabbit antibody (1:200, DyLight 594; Abbkine, Redlands, CA, USA) at room temperature for 1 h. Finally, nuclei were counterstained with DAPI (Beyotime, Shanghai, China) for 5 min. Images were captured with a confocal microscope (Olympus, Tokyo, Japan) and fluorescent quantitative analysis was performed with ImageJ software.

### 2.13. Immunohistochemical Staining

Pulp tissues (four healthy pulps and four inflamed pulps) were obtained from teeth indicated for extraction. Inflamed pulps were obtained from teeth clinically diagnosed with chronic pulpitis. After extraction, teeth were immediately cleft using a chisel and a hammer. Dental pulps were carefully removed with forceps and placed in 4% PFA solution for 24 h. Fixed tissues were embedded in paraffin and sectioned at 4 μm thickness for Hematoxylin-Eosin staining and immunohistochemistry. Sections were deparaffinized and rehydrated, followed by heat-mediated antigen retrieval with immersion in a citrate buffer (pH 8.0) for 10 min at 100 °C in an autoclave. The endogenous peroxidase was quenched with 3% hydrogen peroxide in methanol (20 min at room temperature). To block nonspecific antibody binding, sections were incubated with 10% normal goat serum for 30 min. Subsequently, sections were incubated with antibodies against LOXL2 (1:500, Abcam, Cambridge, UK) overnight in a humid chamber at 4 °C. Concomitantly, negative controls were incubated with PBS instead of the primary antibody. After washing with PBS, a diaminobenzidine (DAB) Detection Kit (Polymer) (Gene Tech, Shanghai, China) was used. The sections were incubated with peroxidase-conjugated secondary antibody for 1 h at room temperature. Reaction products were visualized with DAB chromogen according to the manufacturer’s instructions. A nuclear counter-stain was performed with hematoxylin. Finally, stained slides were dehydrated and mounted. Slides were scanned with a ScanScope slide scanner (Leica, Wetzlar, Germany), and semi-quantitative evaluation was performed using Image J software.

### 2.14. LOXL2 Knockdown in HUVECs

Lentiviral plasmids pLVX-shLOXL2-PURO (Youbio, Changsha, China) containing shRNA targeting the following sequence: 5′-GGAGGACACAGAATGTGAAGGTTCAAGAGACCTTCACATTCTGTGTCCTCCTTTTTT-3′ of LOXL2 were constructed. As a control, pLVX-shControl-PURO lentiviral plasmids containing the nontarget shRNA sequence were used. For lentiviral production, HEK293T cells were transfected with pLVX-shLOXL2-PURO (pLVX-shControl-PURO), pMD2.G and psPAX2 plasmids in a 1:5:10 ratio using 2.5 μL liposomal transfection reagent (Yeasen, Shanghai, China) per μg of DNA. Lentiviral supernatants were harvested after 48 and 72 h. HUVECs were transduced with shLOXL2 or shControl lentivirus for 16 h in the presence of 5 μg/mL polybrene. LOXL2 knockdown was confirmed by quantitative real-time PCR analysis and Western blot analysis.

### 2.15. Statistical Analysis

Statistical analysis was performed using GraphPad Prism 8 (GraphPad, La Jolla, CA, USA) and SPSS 25.0 (IBM Corporation, Armonk, NY, USA). All data are presented as the mean ± standard error from at least 3 independent experiments. Differences between groups were analyzed by Student’s *t*-test for two samples. Comparison among groups was performed by one-way ANOVA. *p* < 0.05 was considered statistically significant.

## 3. Results

### 3.1. Isolation and Characterization of DPSCs

DPSCs isolated from dental pulp tissue were plastic adherent and showed a characteristic fibroblastic spindle-like morphology under a microscope (Figure 1a). Subsequently, we used flow cytometry analysis to demonstrate that the population of dental pulp-derived cells isolated in this study were highly positive for mesenchymal stem cell markers like CD29 (99.14% ± 0.15%), CD73 (99.47% ± 0.41%), CD90 (99.36% ± 0.47) and CD105 (99.40% ± 0.44%), and negative for hematopoietic lineage markers such as CD34 (0.38% ± 0.43%), CD45 (0.35% ± 0.51%) and HLA-DR (0.43% ± 0.27%) (Figure 1b). In addition, under appropriate inductive conditions, DPSCs could differentiate into osteoblasts, adipocytes and endothelial cells (Figure 1c), which indicated that DPSCs exhibit tri-lineage differentiation potential. These results indicated that DPSCs were successfully isolated.

### 3.2. Isolation and Characteristics of Nor-Exos and Hypo-Exos

Nor-Exos and Hypo-Exos were isolated from DPSC conditioned media and hypoxia-preconditioned DPSCs, respectively. TEM and NanoFCM were carried out to identify the harvested DPCs-derived exosomes. To characterize the purified particles as exosomes, the bilayer membrane and “saucer-like” appearance of representative exosomes were examined by TEM (Figure 2a). The NanoFCM analysis showed that the diameters of Nor-Exos and Hypo-Exos mainly ranged from 30 to 150 nm (mean size of 70.59 nm for Nor-Exos and 69.42 nm for Hypo-Exos; Figure 2b). Meanwhile, the Western blot analysis revealed that exosomal surface markers CD9, CD63 and CD81 were expressed in Nor-Exos and Hypo-Exos (Figure 2c). These results indicated the successful extraction of Nor-Exos and Hypo-Exos.

### 3.3. Hypo-Exos Enhance the Angiogenic Capacity of HUVECs In Vitro

Nor-Exos and Hypo-Exos (10 μg/mL) were added to stimulate HUVECs and explore their functional effects on HUVECs. Firstly, to clarify whether Nor-Exos and Hypo-Exos could be taken up by HUVECs, we labeled Nor-Exos and Hypo-Exos with red fluorescent dye PKH26 and incubated them with HUVECs at 37 °C. After 6 h, PKH26-labeled exosomes were taken up by HUVECs, mainly localized in the cytoplasm around the nucleus (Figure 3a). The HUVEC proliferating effects of exosomes were measured by the CCK-8 assay. Results showed that HUVEC proliferation was significantly increased at three days in the presence of either Nor-Exos or Hypo-Exos. Furthermore, HUVECs in the Hypo-Exos group exhibited higher proliferation compared with the Nor-Exos group (Figure 3b). Additionally, the migration of HUVECs was evaluated with a transwell assay. Compared to the control group, the number of migrating cells significantly increased in the Nor-Exos and Hypo-Exos groups. Moreover, Hypo-Exos exhibited a much stronger effect on HUVEC migration than Nor-Exos (Figure 3c,d). The above results suggested that Hypo-Exos exhibited stronger effects on HUVECs proliferation and migration than Nor-Exos in vitro.

To investigate whether hypoxia could enhance the angiogenic capacity of DPSCs-Exos, Matrigel tube formation assays were performed using HUVECs pretreated with PBS, Nor-Exos, or Hypo-Exos 24 h. Both Nor-Exos and Hypo-Exos promoted greater tube formation of HUVECs at 6 h when compared to the control group (Figure 4a). Meanwhile, the stimulation of Hypo-Exos resulted in more junctions and longer tubes than Nor-Exos stimulation. To explore the potential mechanism through which Hypo-Exos promote angiogenesis, we detected the expression of several angiogenesis regulatory molecules in HUVECs, including VEGFA/KDR, SDF-1/CXCR4, CD31 and MMP9. The stimulation of Nor-Exos or Hypo-Exos for 24 h upregulated expression of most angiogenesis associated factors in HUVECs (Figure 4b–d). Interestingly, Hypo-Exos significantly upregulated mRNA and protein levels of VEGFA (*p* < 0.05) rather than VEGFA receptor 2 (KDR) compared with Nor-Exos (Figure 4b–d). Similarly, Hypo-Exos significantly improved protein expression of SDF-1 (*p* < 0.05) rather than its receptor CXCR4 (Figure 4b–d). In addition, the protein expression of CD31 was upregulated (*p* < 0.05) although the mRNA upregulation was not significant compared with the Nor-Exos group (Figure 4b–d). These results suggested that Hypo-Exos may enhance HUVEC angiogenesis by elevating the expression of several proangiogenic factors, especially VEGFA and SDF-1.

### 3.4. Exosomal Protein Identification via iTRAQ-Based Proteomics Analysis

In order to examine the proteomics of Hypo-Exos and the mechanism underlying the angiogenic effect of DPSC-Exos, the two types of exosomes were subjected to iTRAQ-based proteomics analysis. 7114 peptides and 1792 proteins were identified from the Hypo-Exos samples with a 1% FDR. Among these, 1506 proteins overlapped with those reported in the ExoCarta database (http://exocarta.org/, accessed on 1 July 2021) (Figure 5a). Protein enrichment by cellular component analysis identified that the proteins were most enriched in the cell and cell part, while a molecular function analysis revealed that the most abundant proteins were responsible for catalytic and binding activity. Protein enrichment based on biological process analysis indicated that the majority of Hypo-Exos proteins were implicated in cellular and single-organism processes (Figure 5b).

Proteins with an average ratio of more than 1.2 fold and a *p*-value less than 0.05 were determined to be differentially expressed. Proteins differentially-expressed were shown as the volcano map Figure 5c. Compared with the control group, 39 proteins were upregulated and 40 proteins were downregulated in Hypo-Exos (Table 1). The results indicated that the upregulated DEPs include TSNAXIP1, NPM1, BGN, DHX38, COL12A1, SVEP1, LOXL2, FBN1, DCN, FBN2, and SDC4. The downregulated DEPs included PDE3A and MKI67. The primary biological process that DEPs participated in were chondroitin sulfate metabolic and chondroitin sulfate proteoglycan metabolic processes, bone trabecular formation and morphogenesis, angiogenesis, regulation of the fibroblast growth factor receptor signaling pathway, endothelial cell proliferation, regulation of focal adhesion assembly and blood vessel morphogenesis (Figure 5d). Among the proteins involved in the biological process of angiogenesis, THBS1, NCL, ATP5B, MMP-2, HSPG2, TGFBI, and LOXL2 were upregulated, and GNA13 and NRP1 were downregulated. The DEPs pathway analysis is shown in Figure 5e. We found that the pathway most closely associated with DEPs was “ECM-receptor interaction”.

In order to study relationships between the upregulated DEPs, we analyzed the PPI through STRING analysis (https://string-db.org/, accessed on 3 July 2021); Figure 6, which identified close relationships between BGN, MMP2, FBN1, DCN, HSPG2, THBS2, LOXL1, and LOXL2 DEPs.

### 3.5. LOXL2 May Be One of the Key Molecules in Hypo-Exos-Mediated Angiogenesis

LOXL2 belongs to the lysyl oxidase family, which contributes to collagen and elastin cross-linking in the extracellular matrix by mediating the oxidative deamination of the peptide lysine. LOXL2 is also a regulator of sprouting angiogenesis via collagen IV scaffolding. The previous iTRAQ-based proteomic analysis identified LOXL2 as an upregulated protein in Hypo-Exos. We validated the expression of LOXL2 using Western blot analysis (Figure 7a), which verified that expression patterns were consistent with iTRAQ quantitative proteomics results. As previously reported, LOXL2 was a direct transcriptional target of HIF-1α [26]. We aimed to detect the protein expression of HIF-1α and LOXL2 in DPSCs under normoxia and hypoxia by Western blot and immunofluorescence staining. The protein levels of HIF-1α and LOXL2 were markedly increased after hypoxia preconditioning for 48 h as detected by Western blot (Figure 7b) and immunofluorescence staining (Figure 7c). Interestingly, LOXL2 showed predominantly nuclear localization under normoxic conditions while hypoxic conditions were mainly associated with cytoplasmic LOXL2 localization, as determined by immunofluorescence assays (Figure 7c). In vivo, inflammation usually produces a hypoxic microenvironment in dental pulp. We further analyzed LOXL2 expression and location in healthy and inflamed dental pulps by immunohistochemical staining. The healthy and inflammatory state of dental pulp was confirmed by hematoxylin-eosin staining. The pulp structure was clear in the healthy pulp but disorganized in inflamed pulp, along with the discontinuous odontoblast layer, dilated blood vessels and inflammatory cells around blood vessels (Figure 7d-i,d-ii). Healthy dental pulps exhibited weak LOXL2 staining only in the odontoblast layer and in vascular endothelial cells (Figure 7d-iii,d-v). Positive staining was also observed in odontoblasts and endothelial cells in inflamed dental pulps (Figure 7d-iv,d-vi). More notably, inflamed pulps demonstrated the infiltration of inflammatory cells with strong LOXL2 immunoreactivity (Figure 7d-vii). In general, the immunostaining intensity of LOXL2 was higher in inflamed dental pulps than in healthy dental pulps (Figure 7). The experiments above revealed that LOXL2 expression was upregulated by a hypoxic microenvironment in vivo and in vitro.

It is known that LOXL2 regulates the angiogenesis of endothelial cells (ECs). LOXL2 knockdown in ECs results in decreased sprouting [27] and tube formation in a Matrigel-based angiogenesis assay [28]. In order to analyze the role of Hypo-Exos and LOXL2 in HUVEC angiogenesis, we silenced LOXL2 in HUVECs and determined whether the addition of Hypo-Exos restored in vitro angiogenesis. Quantitative real-time PCR analysis and Western blot analysis confirmed that LOXL2 mRNA and protein expression levels were significantly decreased in shLOXL2-HUVECs (Figure 8a). Obviously, the silencing of LOXL2 significantly reduced the number of junctions and total tube length in the Matrigel-based tube formation assay (Figure 8b). Moreover, treatment with Hypo-Exos partially rescued the inhibitory influence of LOXL2 silencing on tube formation in HUVECs (Figure 8b). Altogether, these results partially confirmed that LOXL2 may be one of the key molecules in Hypo-Exos-mediated angiogenesis.

## 4. Discussion

Hypoxia is considered to be a driving force of angiogenesis in injured dental pulp tissue [13]. Hypoxia has been reported to enhance proliferation, migration, differentiation potential and paracrine action of DPSCs [29]. Accumulating evidence suggests that mesenchymal stem cell (MSC)-derived exosomes exhibit stronger angiogenic effects under hypoxic conditions [18,19,20,21,22]. However, exosomes derived from hypoxia-preconditioned DPSCs (Hypo-Exos) have not been characterized in detail and their angiogenic role remains unresolved. In the current study, we observed an induction of HIF-1α in DPSCs exposed to hypoxia. Hypo-Exos demonstrated a stronger effect on HUVEC proliferation, migration and tube formation in vitro, compared to Nor-Exos or the control groups. Furthermore, a comprehensive characterization of Hypo-Exos was obtained through proteomics analysis, which revealed that hypoxia partially alters the proteome profile of DPSC-derived exosomes. Finally, we identified LOXL2 as an upregulated protein in Hypo-Exos and that it was possibly responsible for Hypo-Exos mediated angiogenesis.

DPSCs are mesenchymal stem cells isolated from adult dental pulp. Stem cells have been described to localize to low oxygen tension microenvironments [30], whereas cell culture is usually performed at atmospheric O_2_ concentrations (20–21%). Thus, mimicking the DPSC hypoxic microenvironment may better recapitulate their original properties and promote optimal regenerative responses [31]. Recently, hypoxia preconditioning of stem cells has emerged as a popular strategy to improve stem cell performance in regenerative medicine. DPSCs exhibit better morphology and stemness and stronger migration and proliferation ability under hypoxia as described by Ahmed et al. [29]. More importantly, the secretome profile was significantly affected by hypoxia in a manner that would improve the angiogenic potential. Although it has been reported that conditioned media from hypoxic-cultured human dental pulp cells enhanced angiogenesis [17,32], whether hypoxia preconditioning could enhance the angiogenic potential of exosomes from DPSCs was still unknown before the current report. We previously found that exosomes from dental pulp cells promoted HUVEC proliferation, proangiogenic factor expression and tube formation [5]. In this study, we conducted a series of in vitro experiments to compare the angiogenic potential of Nor-Exos and Hypo-Exos. EC migration and tube formation are two important steps involved in sprouting angiogenesis. The current results indicated that Hypo-Exos enhancement of HUVEC proliferation, migration and tube formation in vitro was superior to that of Nor-Exos. Significant differences in HUVEC proliferation were not detected until three days after exosome application, similar to Gao et al. [21] but different from Xian et al. [5], which may be attributed to the initial cell numbers and detection time. Stimulation of Nor-Exos or Hypo-Exos for 24 h upregulated expression of most proangiogenic factors in HUVECs. Vascular endothelial growth factor (VEGF) and VEGF receptors are essential regulators in physiological and pathological angiogenesis. We observed greater increases in mRNA level and protein expression of VEGFA in Hypo-Exos-treated HUVECs compared to those treated with Nor-Exos. Despite no significant difference in KDR expression between the Nor-Exos and Hypo-Exos groups, KDR levels in both groups were higher than in the control group. These results are similar to a previous study which observed significantly increased VEGF/VEGF-R expression in graft tissue treated with exosomes derived from hypoxia-preconditioned ADSCs compared with normoxia groups and control groups [19]. Likewise, Yuan et al. [22] found that exosomes derived from hypoxia-preconditioned BMSCs significantly promoted VEGF expression of HUVECs compared with exosomes derived from normoxic BMSCs. The current evidence revealed that VEGF/VEGF-R signaling may be a key pathway mediating the angiogenic potential of hypoxia MSC-Exos. Additionally, the results showed that Hypo-Exos increased SDF-1 and CXCR4 expression levels, indicating that the SDF-1/CXCR4 axis may be involved in the angiogenic effects of Hypo-Exos. Actually, the SDF-1/CXCR4 axis has been verified to be important in ECs migration and angiogenesis. Hypoxia could upregulate SDF-1 expression and, on the other hand, promote the sensitivity of HUVECs to SDF-1 by directly or indirectly inducing CXCR4 expression [33]. Moreover, exosomes derived from hypoxia-preconditioned ADSCs were confirmed to possess a higher angiogenesis-enhancing capacity compared to a normoxic group, an effect which was likely mediated via the PKA signaling pathway [18]. A recent study revealed that exosomes derived from hypoxia-preconditioned BMSCs activated the JNK pathway via exosomal HMGB1 and consequently enhanced HUVEC angiogenesis [21]. Altogether, our study suggested that hypoxia preconditioning may be a feasible approach to improve the angiogenic potential of DPSCs exosomes, which would be a promising strategy in cell-free pulp regeneration.

Furthermore, to investigate the mechanism underlying the angiogenic effect of Hypo-Exos, we performed an iTRAQ-based comparative proteomics analysis of the two types of exosomes. Although a proteomics analysis of hypoxia-preconditioned DPSCs has been reported [15], the current report represents the first proteomic study on exosomes from hypoxia-preconditioned DPSCs. A recent study [15] compared the proteome from 3D cultures of human DPSCs under normoxic and hypoxic conditions, and revealed that a portion of the proteins upregulated after hypoxia preconditioning were involved in angiogenesis. Exosomes usually represent the biological characteristics of their parent cells and exhibit effects similar to their parent cells in regeneration medicine [34]. In this study, we found that hypoxia preconditioning partly affected the profile of DPSC-derived exosomes. Compared with the normoxic group, 39 proteins were upregulated and 40 proteins were downregulated in Hypo-Exos. The differentially expressed proteins are mainly involved in chondroitin sulfate and chondroitin sulfate proteoglycan metabolic processes, bone trabecular formation and morphogenesis and angiogenesis, supporting the hypothesis that hypoxia can modulate the proteome profile of DPSCs-derived exosomes and enhance angiogenesis. DPEs such as LOXL2, MMP-2, TGFBI, THBS1, NCL, ATP5B and HSPG2 were involved in the biological process of angiogenesis. Among these proteins, LOXL2, MMP-2, TGFBI, THBS1, NCL, ATP5B and HSPG2 were up-regulated and GNA13 and NRP1 were down-regulated. PPI results showed that LOXL2 and the proteins MMP-2, LOXL1, PCOLCE, FBN1, PLOD1 were closely linked.

LOXL2 belongs to the lysyl oxidase family, which can catalyze the deamination of lysines and hydroxylysines, and promote the cross-linking of elastin and collagen in the extracellular matrix. 72 kDa type IV collagenase (Matrix metalloproteinase-2, MMP2) participated in the degradation of the ECM and are involved in diverse functions such as vasculature remodeling, angiogenesis, tissue repair, tumor invasion and inflammation. It was found that exosomal MMP2 derived from mature osteoblasts promotes the angiogenesis of ECs in vitro through the VEGF/Erk1/2 signaling pathway [35]. Transforming growth factor-beta-induced protein ig-h3 (TGFBI) is an extracellular matrix molecule involved in various aspects of tumorigenesis, including tumor progression, angiogenesis and metastasis. Thrombospondin-1 (THBS1) plays a role in collagen homeostasis through interactions with matrix metalloproteinases and TGFBI [36]. Interestingly, LOXL2 inhibition significantly suppressed MMP2 and TGFBI expression [37,38]. Another protein which is upregulated in Hypo-Exos is Procollagen-lysine, 2-oxoglutarate 5-dioxygenase 1(PLOD1). PLOD1 is involved in collagen synthesis, cross-linking and deposition, and regulates cell proliferation, migration, invasion and apoptosis. It has already been observed that PLOD1 mutations and overexpression promote the occurrence and metastasis of malignant tumors [39]. Moreover, HIF-1 can strongly promote PLOD1 transcription and expression under a hypoxic environment, which is consistent with our proteomic results. Fibrillin 1 (FBN1) is an important protein of the extracellular matrix, and contributes to the final structure of a microfibril. Procollagen C-endopeptidase enhancer 1 (PCOLCE) is known to play a crucial role in procollagen maturation and collagen fibril formation. Since LOXL2 correlated with several genes involved in collagen remodeling, including MMP2, TGFBI [40], thrombospondin-1 (THBS1) [36], PCOLCE [41] and FBN1 [42], and has been shown to take part in the biological process of angiogenesis, it would be of interest to evaluate whether exosomal LOXL2 promote angiogenesis.

Hypoxia induces the expression and secretion of LOXL2, which participates in capillary formation through EC proliferation and migration and collagen IV network assembly [43]. Similarly, in our study, hypoxia preconditioning upregulated the expression of HIF-1α and LOXL2 in DPSCs. Others reported that hypoxia-induced LOXL2 upregulation was mediated by HIF-1α, and a regulatory loop between LOXL2 and HIF-1α was further confirmed [26,44,45]. After hypoxia treatment, the nucleo-cytoplasmic transfer occurred on the expression of LOXL2, indicating the increased secretion of LOXL2 into the extracellular, which is consistent with our previous proteomics result that LOXL2 was up-regulated in hypoxia DPSC-Exos. Inflammation is known to cause ischemia and hypoxia in the dental pulp. Therefore, we investigated and compared for the first time the expression and distribution of LOXL2 in the healthy and inflamed dental pulp. LOXL2 was mildly expressed in the odontoblast layer and vascular ECs and strongly expressed in inflamed regions, especially in inflammatory cells. These expression patterns suggest that LOXL2 may play an essential role in the hypoxic pulp. In this study, we observed that hypoxia not only promoted the expression of LOXL2 in DPSCs, but also increased LOXL2 levels in DPSC-Exos as determined by proteome analysis and Western blot analysis. Considering that LOXL2 is an important regulatory molecule in angiogenesis, as evidenced by reduced sprouting angiogenesis [27] and tube formation [28] after LOXL2 knockdown, we supposed that LOXL2 may be one of the key molecules in Hypo-Exos-mediated angiogenesis. To test this, we silenced LOXL2 in HUVECs and carried out a rescue experiment with Hypo-Exos. Silencing LOXL2 dramatically inhibited tube formation, and this was partially rescued by Hypo-Exos treatment. The above evidence supports our hypothesis in part. Experiments utilizing functional gain and loss of LOXL2 in Hypo-Exos are needed to further investigate whether Hypo-Exos enhances angiogenesis by transferring LOXL2.

At present, there are still some limitations in the definition of exosomes. According to the guidelines of the International Society for Extracellular Vesicles [46], the term “EVs” was advised to replace “exosomes”, since it is difficult to distinguish endosome-origin “exosomes” from other extracellular vesicles (EVs) because of the overlapping size range and the lack of specific markers. In our study, we characterized exosomes by detecting the morphology, particle size and protein markers CD9, CD63 and CD81. However, there was a lack of direct evidence tracking the release of exosomes. Indeed, the biogenesis of exosomes was considered to differ from other extracellular vesicles [47,48]. Considering that the isolation method and particle size in our study were similar to many relevant studies [3,6,8], we described these particles as “exosomes” here. MSC exosomes contain complex and diverse miRNAs that play an important role in MSC-mediated angiogenesis [49]. For instance, exosomal miR-126 was confirmed to mediate the angiogenic effect of hypoxia-preconditioned human umbilical cord mesenchymal stem cells [20]. Both proteins and miRNAs could mediate the angiogenic effect of Hypo-Exos. However, we focused only on the protein cargo of DPSC-derived exosomes in the present study. Based on the evidence available, we selected LOXL2 as a candidate molecule and subsequently will investigate the impact of DPSC exosomal LOXL2 on angiogenesis.

## 5. Conclusions

In conclusion, we demonstrated that both Nor-Exos and Hypo-Exos enhanced HUVEC proliferation, migration and tube formation in vitro. Moreover, Hypo-Exos exhibited stronger angiogenic effects than Nor-Exos. Furthermore, iTRAQ-based proteomics analysis revealed that hypoxia partially alters the proteome profile of DPSC-derived exosomes. Finally, we proposed that LOXL2 is a candidate molecule through which Hypo-Exos promotes angiogenesis. Further experiments are currently underway to verify this hypothesis.

## Figures and Tables

**Figure 1 biomolecules-12-00575-f001:**
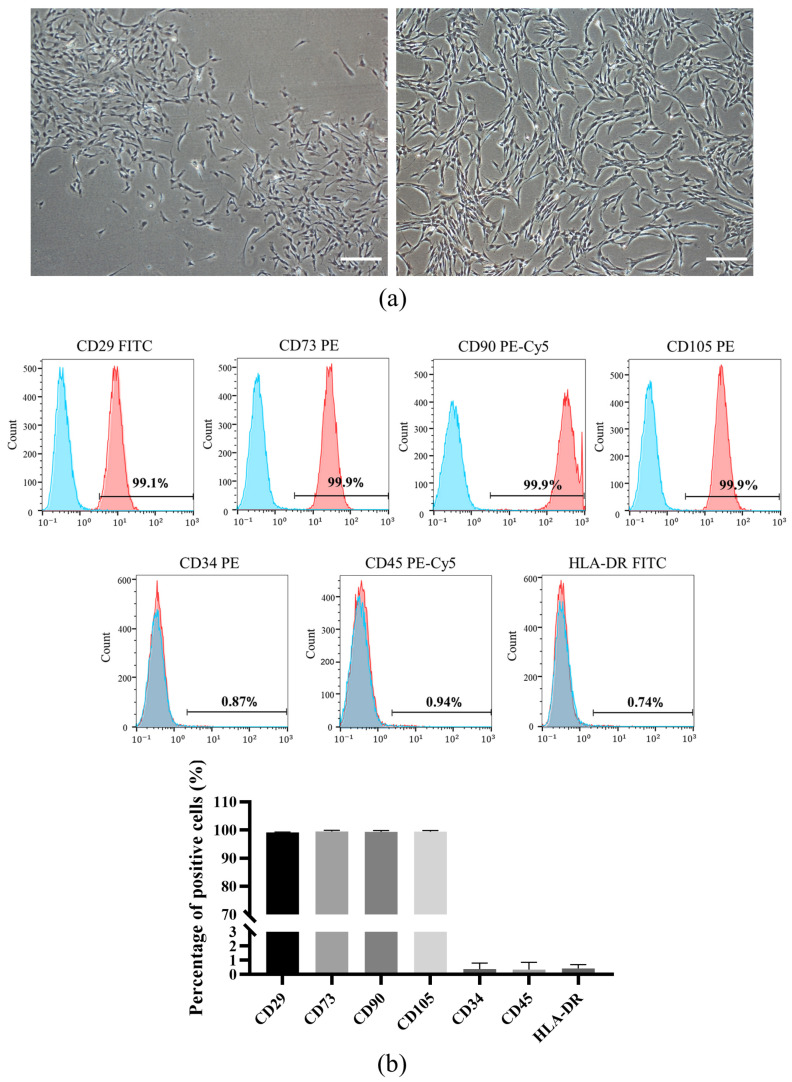
Characterization of DPSCs. (**a**) DPSC morphology was observed under an optical microscope. (**b**) DPSC phenotypes were examined by flow cytometry. (**c**) Osteogenic differentiation, adipogenic differentiation and endothelial differentiation of DPSCs. *n* = 3, mean ± SD. Scale bars: 200 μm (**a**,**c-iii**); 100 μm (**c-i**,**c-ii**).

**Figure 2 biomolecules-12-00575-f002:**
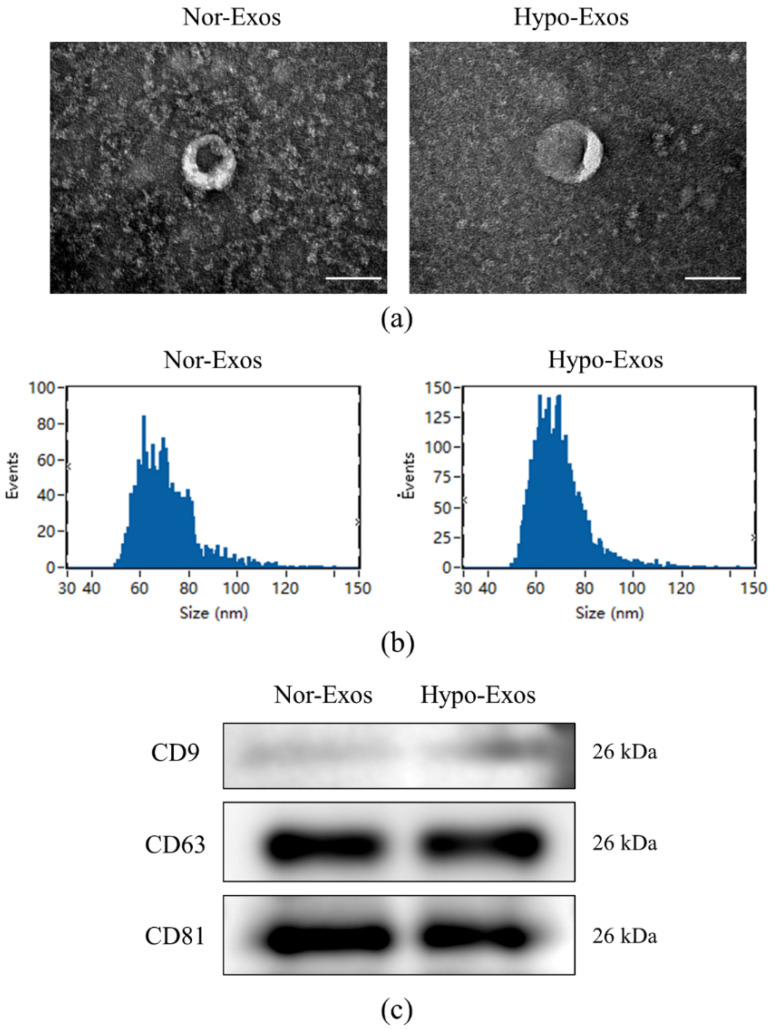
Characterization of Nor-Exos and Hypo-Exos. (**a**) Morphology of Nor-Exos and Hypo-Exos under TEM. (**b**) Diameter distribution of Nor-Exos and Hypo-Exos assessed by NanoFCM. (**c**) Exosomal surface markers CD9, CD63 and CD81 were detected by Western blot analysis. Scale bars: 100 nm (**a**).

**Figure 3 biomolecules-12-00575-f003:**
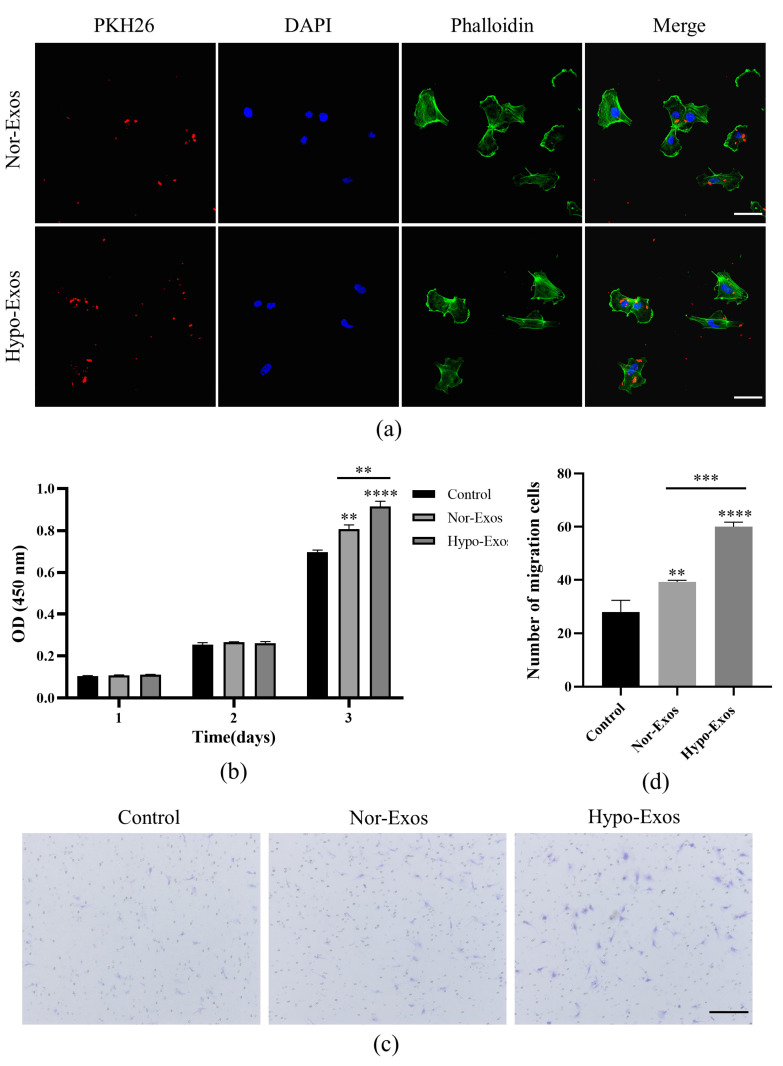
Effects of Nor-Exos and Hypo-Exos (10 μg/mL) on HUVEC proliferation and migration. (**a**) Fluorescence microscopy analysis of PKH26-labelled exosomes (red) internalized by HUVECs. The cytoskeleton was stained by phalloidin (green) and nuclei were stained with DAPI (blue). (**b**) The proliferation of HUVECs was assessed by CCK-8 assay. (**c**) The migration of HUVECs was evaluated by transwell assays. (**d**) Quantitative analysis of the migrated cells in (**c**). *n* = 3, mean ± SD, ** *p* < 0.01, *** *p* < 0.001, **** *p* < 0.0001. Scale bars: 50 μm (**a**); 200 µm (**c**).

**Figure 4 biomolecules-12-00575-f004:**
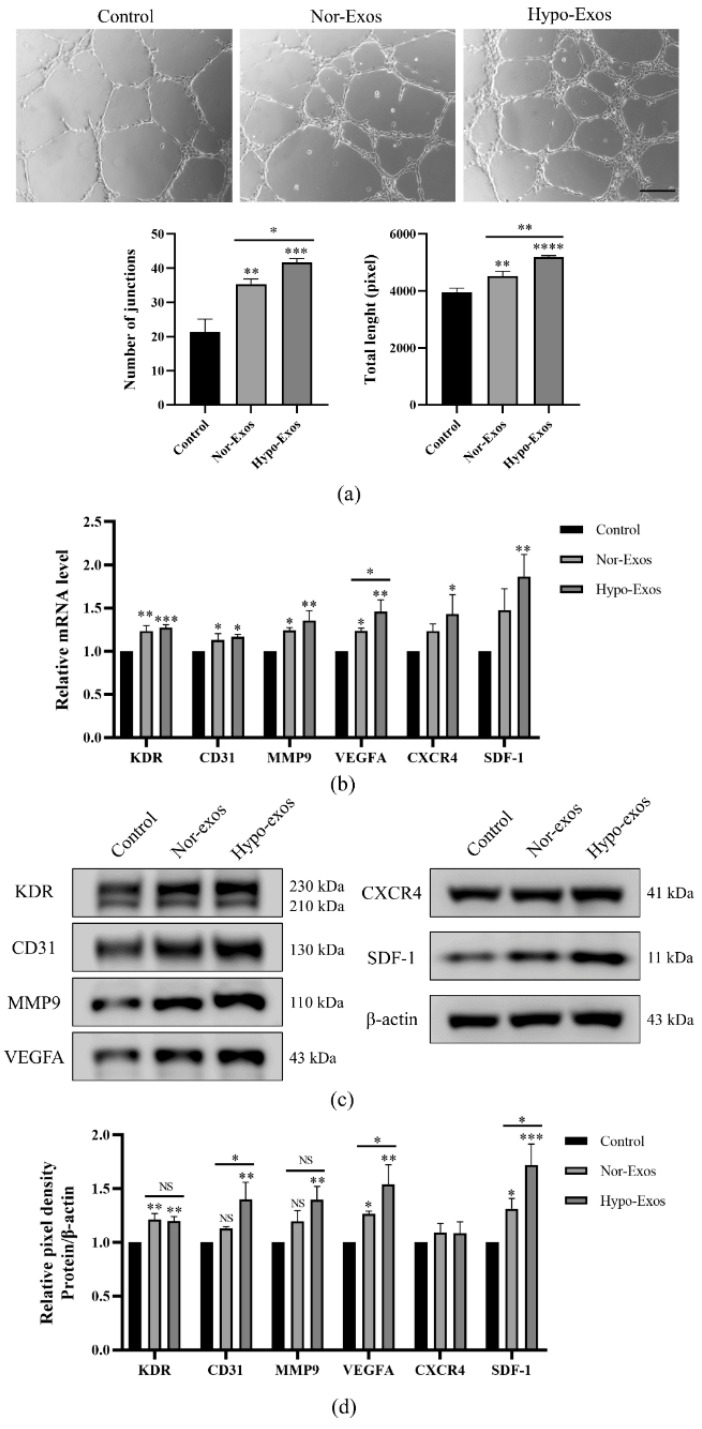
Effects of Nor-Exos and Hypo-Exos (10 μg/mL) on HUVECs in vitro angiogenesis. (**a**) In vitro tube formation of HUVECs. (**b**) The mRNA levels of angiogenesis-associated factors in HUVECs were detected by quantitative real-time PCR. (**c**) The protein levels of angiogenesis-associated factors in HUVECs were detected by Western blot analysis. (**d**) Quantitative analysis of the relative protein expression in (**c**). *n* = 3, mean ± SD, * *p* < 0.05, ** *p* < 0.01, *** *p* < 0.001, **** *p* < 0.0001. NS: no significance. Scale bars: 200 µm (**a**).

**Figure 5 biomolecules-12-00575-f005:**
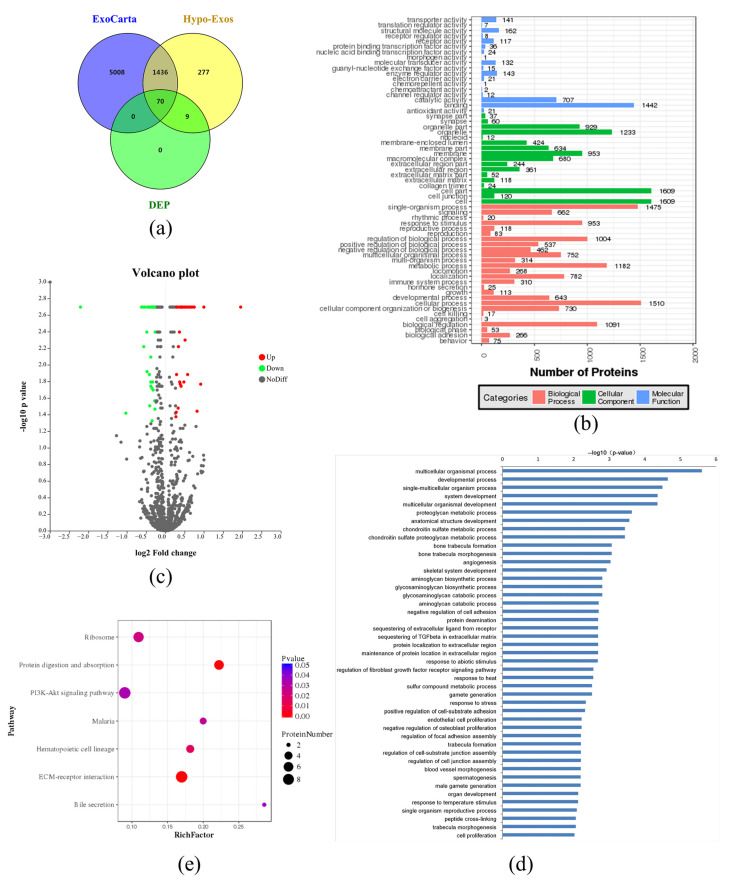
Exosomal protein identification via iTRAQ-based proteomic analysis. (**a**) Venn diagrams showed that 1792 proteins were identified from the Hypo-Exos samples, among which 1506 proteins overlap with that reported in ExoCarta (http://exocarta.org/, accessed on 1 July 2021). Seventy-nine differentially-expressed proteins (DEPs) were identified, and 70 were found in the ExoCarta database. (**b**) GO annotation analysis of 1792 exosomal proteins via Blast2go. Proteins were classified by cellular component, molecular function, and biological process. (**c**) Volcano plot representing the fold change and *p*-value in Hypo-Exos and Nor-Exos. x-Axis showed fold change, while *y*-axis depicted *p*-value. Red for upregulated proteins, green for downregulated proteins, gray for no significant expressed proteins. (**d**) GO Biological process of DEPs in Hypo-Exos vs. Nor-Exos. (**e**) KEGG pathway analysis of DEPs in Hypo-Exos vs. Nor-Exos.

**Figure 6 biomolecules-12-00575-f006:**
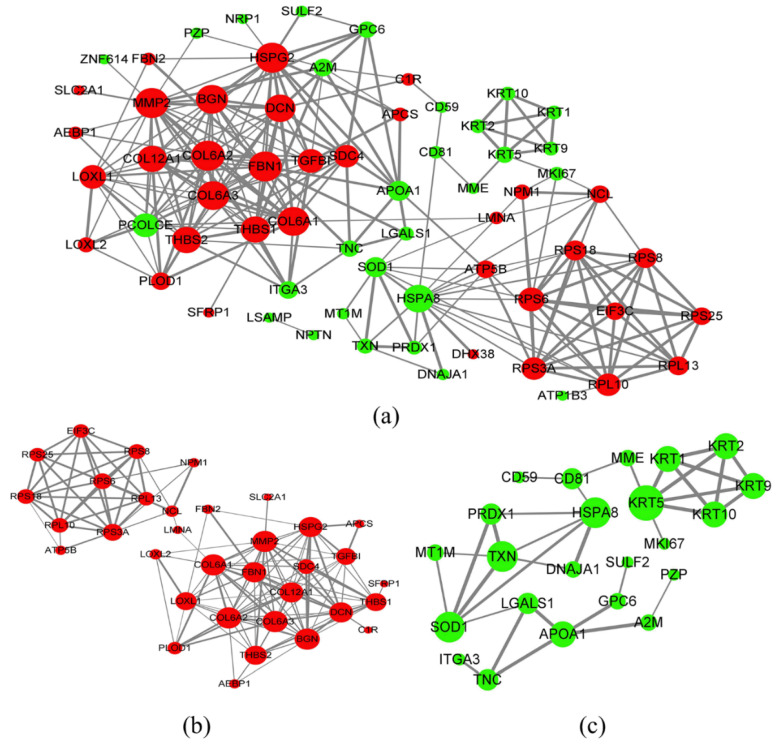
Protein-protein interaction network obtained using STRING and Cytoscape software in differentially-expressed proteins in Hypo-Exos vs. Nor-Exos. Upregulated and downregulated proteins are indicated by the red and green nodes, respectively. The greater the number of interacting proteins, the bigger the nodes. The stronger the data support, the thicker the line. (**a**) Network of proteins obtained in all the DEPs. (**b**) Network of proteins obtained in the upregulated DEPs. (**c**) Network of proteins obtained in the downregulated DEPs.

**Figure 7 biomolecules-12-00575-f007:**
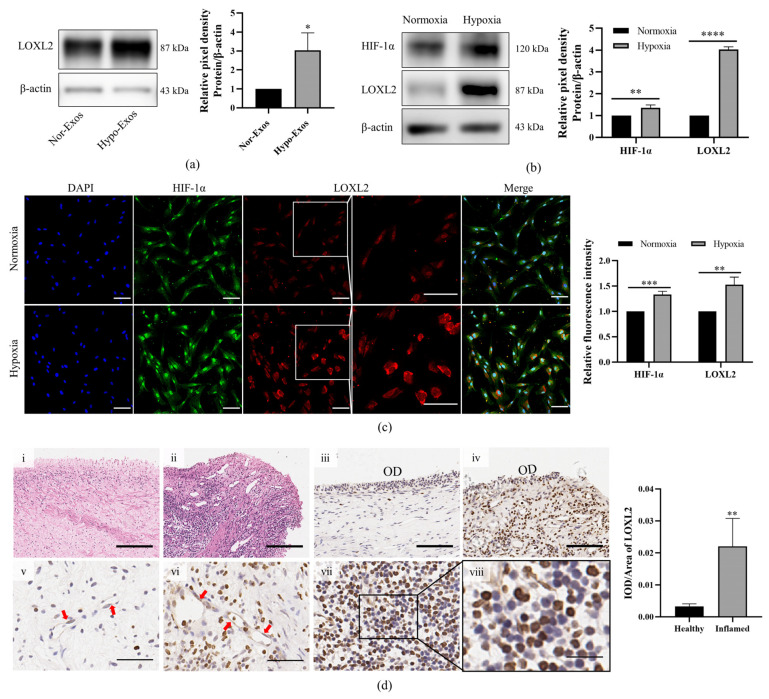
Protein expression of LOXL2 in DPSC-Exos, DPSCs and dental pulp tissue. (**a**) Detection of exosomal protein levels of LOXL2 in normoxia and hypoxia-preconditioned DPSCs by Western blot analysis. (**b**) Protein expression of HIF1α and LOXL2 in DPSCs under normoxia or hypoxia by Western blot. (**c**) Immunofluorescence images of HIF1α (green) and LOXL2 (red) in DPSCs under normoxia or hypoxia. Nuclei were stained with DAPI (blue). (**d**) Immunolocalization of LOXL2 in healthy and inflamed human dental pulp tissues. Healthy (**d-i**) and inflamed (**d-ii**) dental pulp stained with hematoxylin-eosin. Positive staining of LOXL2 in the odontoblast layer (OD) of healthy pulp (**d-iii**) and inflamed pulp (**d-iv**). Positive staining of LOXL2 in endothelial cells (red arrow) of healthy pulp (**d-v**) and inflamed pulp (**d-vi**). (**d-vii**) Intense staining of inflammatory cells in inflamed pulps. *n* = 3 (**a**–**c**), *n* = 4 (**d**), mean ± SD, * *p* < 0.05, ** *p* < 0.01, *** *p* < 0.001, **** *p* < 0.0001. Scale bars: 200 μm (**d-i**,**d-ii**); 100 μm (**c**,**d-iii**,**d-iv**); 50 μm (**d-v**,**d-vi**); 20 μm (**d-viii**).

**Figure 8 biomolecules-12-00575-f008:**
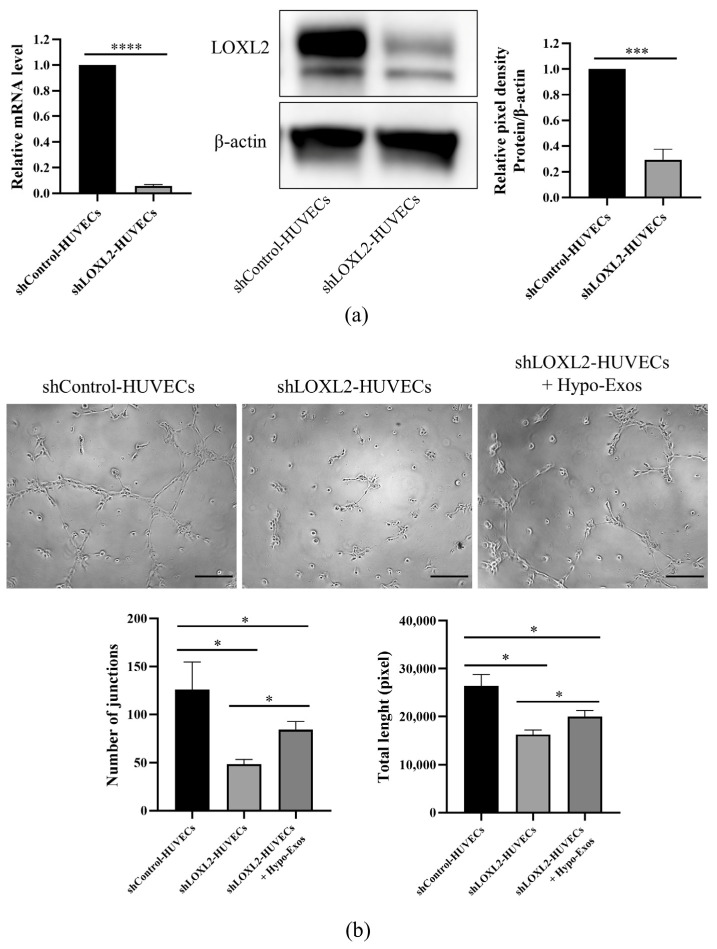
LOXL2 knockdown in HUVECs and tube formation rescue experiment. (**a**) LOXL2 mRNA and protein expression levels were detected by quantitative real-time PCR analysis and Western blot analysis. (**b**) Hypo-Exos partially rescued the angiogenesis of shLOXL2-HUVECs in the Matrigel-based tube formation assay. *n* = 3, mean ± SD, * *p* < 0.05, *** *p* < 0.001, **** *p* < 0.0001. Scale bars: 200 μm (**b**).

**Table 1 biomolecules-12-00575-t001:** 79 proteins were differentially-expressed between Hypo-Exos and Nor-Exos.

Protein ID	Protein Name	Abbreviation	Ratio	*p*-Value	Expression
Q4LDE5	Sushi, von Willebrand factor type A, EGF and pentraxin domain-containing protein 1	SVEP1	1.666	0.002	up
Q99715	Collagen alpha-1(XII) chain	COL12A1	1.675	0.002	up
P07585	Decorin	DCN	1.554	0.002	up
Q8N474	Secreted frizzled-related protein 1	SFRP1	1.283	0.004	up
Q08397	Lysyl oxidase homolog 1	LOXL1	1.311	0.002	up
P07996	Thrombospondin-1	THBS1	1.306	0.002	up
P02743	Serum amyloid *p*-component	APCS	1.325	0.002	up
P06748	Nucleophosmin	NPM1	1.991	0.002	up
P62753	40S ribosomal protein S6	RPS6	1.218	0.013	up
Q9Y383	Putative RNA-binding protein Luc7-like 2	LUC7L2	1.39	0.016	up
P11166	Solute carrier family 2, facilitated glucose transporter member 1	SLC2A1	1.272	0.002	up
Q9Y4K0	Lysyl oxidase homolog 2	LOXL2	1.615	0.002	up
P31431	Syndecan-4	SDC4	1.487	0.013	up
P62269	40S ribosomal protein S18	RPS18	1.211	0.037	up
P26373	60S ribosomal protein L13	RPL13	1.302	0.017	up
P12110	Collagen alpha-2(VI) chain	COL6A2	1.371	0.002	up
Q8IUX7	Adipocyte enhancer-binding protein 1	AEBP1	1.335	0.002	up
Q99613	Eukaryotic translation initiation factor 3 subunit C	EIF3C	1.289	0.016	up
P27635	60S ribosomal protein L10	RPL10	1.321	0.018	up
P00736	Complement C1r subcomponent	C1R	1.317	0.002	up
Q15582	Transforming growth factor-beta-induced protein ig-h3	TGFBI	1.459	0.002	up
P35555	Fibrillin-1	FBN1	1.595	0.002	up
P19338	Nucleolin	NCL	1.343	0.002	up
Q2TAA8	Translin-associated factor X-interacting protein 1	TSNAXIP1	3.862	0.002	up
P61247	40S ribosomal protein S3a	RPS3A	1.26	0.006	up
P02545	Prelamin-A/C	LMNA	1.204	0.042	up
P35442	Thrombospondin-2	THBS2	1.316	0.002	up
Q02809	Procollagen-lysine,2-oxoglutarate 5-dioxygenase 1	PLOD1	1.216	0.002	up
P08253	72 kDa type IV collagenase	MMP2	1.399	0.002	up
Q92620	Pre-mRNA-splicing factor ATP-dependent RNA helicase PRP16	DHX38	1.762	0.036	up
P06576	ATP synthase subunit beta, mitochondrial	ATP5B	1.221	0.002	up
P35556	Fibrillin-2	FBN2	1.513	0.002	up
P21810	Biglycan	BGN	1.881	0.017	up
Q96KK5	Histone H2A type 1-H	HIST1H2AH	1.253	0.033	up
P62851	40S ribosomal protein S25	RPS25	1.42	0.005	up
P62241	40S ribosomal protein S8	RPS8	1.392	0.002	up
P12111	Collagen alpha-3(VI) chain	COL6A3	1.465	0.002	up
P98160	Basement membrane-specific heparan sulfate proteoglycan core protein	HSPG2	1.234	0.002	up
P12109	Collagen alpha-1(VI) chain	COL6A1	1.418	0.002	up
P35908	Keratin, type II cytoskeletal 2 epidermal	KRT2	0.725	0.002	down
P46013	Proliferation marker protein Ki-67	MKI67	0.489	0.038	down
P31689	DnaJ homolog subfamily A member 1	DNAJA1	0.822	0.027	down
P08473	Neprilysin	MME	0.757	0.002	down
P09382	Galectin-1	LGALS1	0.797	0.002	down
P01591	Immunoglobulin J chain	JCHAIN	0.713	0.004	down
P0DOX8	Immunoglobulin lambda-1 light chain	IGL1	0.768	0.002	down
P02647	Apolipoprotein A-I	APOA1	0.812	0.002	down
P13645	Keratin, type I cytoskeletal 10	KRT10	0.663	0.002	down
P20742	Pregnancy zone protein	PZP	0.767	0.018	down
P13647	Keratin, type II cytoskeletal 5	KRT5	0.784	0.002	down
O14786	Neuropilin-1	NRP1	0.807	0.002	down
P00441	Superoxide dismutase [Cu-Zn]	SOD1	0.825	0.034	down
P24821	Tenascin	TNC	0.755	0.002	down
Q9Y625	Glypican-6	GPC6	0.766	0.008	down
P29144	Tripeptidyl-peptidase 2	TPP2	0.675	0.006	down
Q14432	cGMP-inhibited 3′,5′-cyclic phosphodiesterase A	PDE3A	0.216	0.002	down
P04264	Keratin, type II cytoskeletal 1	KRT1	0.684	0.002	down
P13987	CD59 glycoprotein	CD59	0.781	0.002	down
Q14344	Guanine nucleotide-binding protein subunit alpha-13	GNA13	0.799	0.02	down
P69891	Hemoglobin subunit gamma-1	HBG1	0.76	0.002	down
Q8N339	Metallothionein-1M	MT1M	0.716	0.012	down
Q9Y639	Neuroplastin	NPTN	0.812	0.002	down
O43866	CD5 antigen-like	CD5L	0.782	0.019	down
Q9ULC3	Ras-related protein Rab-23	RAB23	0.792	0.016	down
Q99536	Synaptic vesicle membrane protein VAT-1 homolog	VAT1	0.829	0.004	down
Q15113	Procollagen C-endopeptidase enhancer 1	PCOLCE	0.649	0.002	down
Q06830	Peroxiredoxin-1	PRDX1	0.766	0.016	down
P10599	Thioredoxin	TXN	0.723	0.002	down
Q8N883	Zinc finger protein 614	ZNF614	0.807	0.002	down
Q8IWU5	Extracellular sulfatase Sulf-2	SULF2	0.752	0.031	down
Q13449	Limbic system-associated membrane protein	LSAMP	0.784	0.047	down
P04220	Ig mu heavy chain disease protein	IGHM	0.691	0.002	down
P35527	Keratin, type I cytoskeletal 9	KRT9	0.733	0.002	down
O43854	EGF-like repeat and discoidin I-like domain-containing protein 3	EDIL3	0.783	0.002	down
P01023	Alpha-2-macroglobulin	A2M	0.779	0.002	down
P54709	Sodium/potassium-transporting ATPase subunit beta-3	ATP1B3	0.747	0.013	down
P26006	Integrin alpha-3	ITGA3	0.81	0.002	down
P60033	CD81 antigen	CD81	0.758	0.002	down
P11142	Heat shock cognate 71 kDa protein	HSPA8	0.828	0.002	down

## Data Availability

The mass spectrometry proteomics data have been deposited to the ProteomeXchange Consortium via the PRIDE partner repository with the dataset identifier PXD032719.

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
