# Peer review of "Hypoxia Alters the Proteome Profile and Enhances the Angiogenic Potential of Dental Pulp Stem Cell-Derived Exosomes"

_biomolecules, 2022, doi:10.3390/biom12040575_

Round 1

Reviewer 1 Report

  1. To clearly show the cell morphology, Figure1a should be replaced by higher-amplification images.
  2. A statistical analysis should be added to the flow data of Figure1b.
  3. It would be better for the data presentation if semi-quantitative analysis is added to the staining data in Figure 1c and Figure 7d.
  4. A statistical analysis should be added to Figure 2c. 
  5. Figure 3a should be adjusted to higher brightness for better visualization.
  6. For the exosome cell treatment, how was the does (10ug/ml) defined? Please illustrate this in Methods.
  7. Was the proteomics analysis performed with biological replicates (DPSCs-exosomes isolated from different individuals)? Please clarify this in the Methods. If so, a PCA or UMAP plot should be included in Figure 5 to show the variation between samples.
  8. Higher-amplification confocal images should be used to replace Figure 7c.

Reviewer 2 Report

The manuscript, “Hypoxia alters the proteome profile and enhances the angiogenic potential of dental pulp stem cell-derived exosomes” is a well presented and interesting manuscript. However, the novelty of the concept is questionable. Following are a few questions that need clarification.

  1. Authors have mentioned, “Exosomes derived from different types of MSCs differ in composition and function…….” Please provide references for this sentence.
  2. There are manuscripts that demonstrated the angiogenic properties of exosomes derived from hypoxia-induced mesenchymal stem cells. For instance, “Exosomes Secreted from Hypoxia-Preconditioned Mesenchymal Stem Cells Prevent Steroid-Induced Osteonecrosis of the Femoral Head by Promoting Angiogenesis in Rats” by Li et al., Please include findings from these types on studies in the introduction as well as in discussion.
  3. What is the rationale of using dilution factor for exosomes as 1:9?
  4. What is the reason for using 10 μg/mL of Nor-Exos and Hypo-Exos in experiments?
  5. For western blotting and iTRAQ Labeling indicate how the samples were prepared.
  6. In Figure 8, what is the reason for missing Nor-Exos group?
  7. In the discussion, authors have mentioned, “whether hypoxia preconditioning could enhance the angiogenic potential of exosomes from DPSCs the same as ADSCs and BMSCs was still unknown before the current report.” But in the current report, the authors are not comparing ADSCs or BMSCs.

Round 2

Reviewer 1 Report

Accept in present form

Author Response

Thanks for your careful reading and acceptance of our responses and the revised manuscript. Wish you have a nice day.